# HOFT: Householder Orthogonal Fine-tuning

## Abstract

Adaptation of foundation models using low-rank methods is a widespread approach. Another way to adapt these models is to employ orthogonal fine-tuning methods, which are less time and memory efficient despite their good generalization properties. In this work, we propose Householder Orthogonal Fine-tuning (HOFT), a novel orthogonal fine-tuning method that aims to alleviate the time and space complexity. Moreover, some theoretical properties of the orthogonal fine-tuning paradigm are explored. From this exploration, Scaled Householder Orthogonal Fine-tuning (SHOFT) is proposed. Both HOFT and SHOFT are evaluated in downstream tasks, namely commonsense reasoning, machine translation, subject-driven generation and mathematical reasoning. Compared with state-of-the-art adaptation methods, HOFT and SHOFT show comparable or better results.

## 1 Introduction

Nowadays, fine-tuning foundation models for downstream tasks [17] is the standard approach to model adaptation these days thanks to their knowledge across many domains. By tuning far fewer parameters than full fine-tuning using parameter-efficient fine-tuning techniques (PEFT), the model is able to learn the key aspects of a task and perform comparably or even better than full fine-tuning [18]. This fact makes PEFT methods a particularly well-suited approach for efficiently adapting these models. The employment of PEFT methods has enabled the adaptation of large foundation models without the necessity of compute-intensive hardware infrastructure, making adaptation accessible to a broader user community.

The most popular PEFT methods are based on low-rank approximations, including Low-Rank Adaptation (LoRA) [19] and Weight-Decomposed Low-Rank Adaptation (DoRA) [27]. These methods work under the assumption that the learnable parameters must reside in a lower intrinsic dimension [2]. Alternatively, there are methods proposing the use of orthogonal matrices for adaptation, such as Orthogonal Fine-tuning (OFT) [36] and Orthogonal Butterfly (BOFT) [29]. These methods hypothesize that a good fine-tuned model should have a minimal difference in hyperspherical energy compared to the pre-trained model [26, 28]. In brief, the assumption made is that orthogonality is required to learn new features while keeping pre-trained information [36, 53]. Whilst the performance of these techniques has been demonstrated, their runtime and memory footprint make them a less preferable option for use in real-world applications. A recent approach to balance low-rank and orthogonal methods is Householder Reflection Adaption (HRA) [53], which constrains orthogonality through the incorporation of a term within the loss function. With the employment of an additional weight $\lambda$ for the orthogonality regularizer, HRA aims to construct the chained matrix product of Householder transformations [14].

Orthogonal fine-tuning methods generally result in the construction of a single orthogonal matrix for adaptation purposes. This work demonstrates that two orthogonal matrices are required in order to ensure full expressivity in orthogonal fine-tuning methods. This leads us to propose **Householder Orthogonal Fine-tuning** (HOFT): a novel orthogonal fine-tuning technique using two orthogonal

matrices as directional components efficiently updated through orthogonal transformations. For efficiency, these matrices are obtained by accumulating Householder transformations via the CWY transform [25, 21] along with a fast inverse approximation. Additionally, we draw inspiration from DoRA's analysis, which shows that fine-tuning magnitude and direction separately closely matches the learning dynamics of full fine-tuning. From this, a variant of HOFT incorporating an additional scaling transformation is proposed: **Scaled Householder Orthogonal Fine-tuning** (SHOFT).

In order to evaluate both methods, a series of experiments are conducted in four distinct areas: commonsense reasoning, machine translation, subject-driven generation and mathematical reasoning. The selection of these tasks was made with the intention of evaluating the efficacy of the proposed methods along with both low-rank and orthogonal PEFT. Notably, quantized models are also adapted in mathematical reasoning experiments. Experimental results demonstrate that HOFT and SHOFT benefit from retaining the relational structure of pre-trained weights, reaching or exceeding the performance of existing state-of-the-art PEFT baselines.

## 2   Related Work

**Low-Rank Adaptation**   Methods in this family assume that effective fine-tuning updates lie on a compact, low-dimensional manifold [19, 27, 23, 55, 24, 51, 20, 46]. LoRA [19] introduces trainable low-rank adapter matrices into each Transformer layer, freezing the original weights and reducing trainable parameters by several orders of magnitude. DoRA [27] separates the fine-tuning of directional and scaling components by normalizing LoRA's output and applying a scaling transformation. PiSSA [30] employs singular value decomposition (SVD) on pre-trained weight matrices to initialize LoRA adapters in principal subspaces, maintaining most of the original model's expressive capacity. QLoRA [12] combines 4-bit NormalFloat (NF4) quantization with LoRA, enabling the fine-tuning of 65B-parameter models on a single 48GB GPU while preserving near full-precision quality. QA-LoRA [49] employs group-wise quantization operators to selectively compress adapter updates with minimal impact on task performance loss.

**Orthogonal Fine-Tuning**   Orthogonal fine-tuning methods learn distance preserving transformations in weight space, keeping geometric properties such as hyperspherical energy among neuron activations [36, 29]. Previous works show how the imposition of orthogonality constraints within deep learning architectures is conducive to enhancing performance [5, 44, 48, 13, 3]. OFT [36] employs Cayley parameterization [22] to generate orthogonal matrix blocks. Additionally, COFT [36] constrains the orthogonal matrix to be within a small neighborhood of the pre-trained matrix. BOFT [29] reduces OFT parameter footprint by factorizing orthogonal updates into butterfly structures inspired by the Cooley–Tukey FFT algorithm [6], achieving similar generalization gains with fewer trainable parameters. The employment of hybrid methods, such as HRA, enforces hyperspherical constraints on low-rank adapters to blend both paradigms via a term in the loss function controlled by a weight [53].

## 3   Proposed Method

### 3.1   Orthogonal fine-tuning paradigm

As discussed in Section 2, orthogonal fine-tuning stresses the importance of preserving the hyperspherical energy of the given matrix $\mathbf{M} = \mathbf{U}\mathbf{\Sigma}\mathbf{V}^\top \in \mathbb{R}^{m \times n}$. Although it is clear that this can be done by adapting both singular vector matrices $\mathbf{U}$ and $\mathbf{V}$, it is common practice to keep $\mathbf{V}^\top$ unchanged and adapt only $\mathbf{U}$ [36, 29, 53].

Consider all possible orthogonal transformations of $\mathbf{M}$ into an adapted matrix $\widehat{\mathbf{M}} = \widehat{\mathbf{U}}\widehat{\mathbf{\Sigma}}\widehat{\mathbf{V}}^\top$ preserving its hyperspherical energy; that is, meaning that $\widehat{\mathbf{\Sigma}} = \mathbf{\Sigma}$, though $\widehat{\mathbf{U}}$ and $\widehat{\mathbf{V}}^\top$ might differ from $\mathbf{U}$ and $\mathbf{V}^\top$ respectively. Suppose there exists an orthogonal matrix $\mathbf{Q} \in \mathrm{O}(m)$ such that $\widehat{\mathbf{M}} = \mathbf{Q}\mathbf{M}$, that is $\widehat{\mathbf{U}}\widehat{\mathbf{\Sigma}}\widehat{\mathbf{V}}^\top = \mathbf{Q}\mathbf{U}\mathbf{\Sigma}\mathbf{V}^\top$. Since $\mathbf{Q}$ is arbitrary, we can set $\mathbf{Q} = \widehat{\mathbf{U}}\mathbf{U}^\top$, and due to hyperspherical energy conservation, $\widehat{\mathbf{\Sigma}} = \mathbf{\Sigma}$. However, we cannot ensure that $\mathbf{V}$ and $\widehat{\mathbf{V}}$ are equal. Thus, in order to cover all possible adapted matrices, we need two orthogonal matrices $\mathbf{Q_U} \in \mathrm{O}(m), \mathbf{Q_V} \in \mathrm{O}(n)$. Only in this case we can ensure that it is possible to obtain $\widehat{\mathbf{M}}$, since we can set $\mathbf{Q_U} = \widehat{\mathbf{U}}\mathbf{U}^\top$ and $\mathbf{Q_V} = \mathbf{V}\widehat{\mathbf{V}}^\top$ to construct $\mathbf{Q_U}\mathbf{M}\mathbf{Q_V} = \widehat{\mathbf{M}}$.

In terms of approximation error, pre- and post-multiplying the pre-trained matrix $\mathbf{M}$ by distance-preserving transformations exactly captures all adapted matrices $\widehat{\mathbf{M}}$ that maintain the same hyper-spherical energy. However, the error incurred when applying just one orthogonal matrix leads us to a known problem, the Orthogonal Procrustes Problem [15], which has a solution if $\widehat{\mathbf{M}}$ and $\mathbf{M}$ are known matrices. In this one-transform setting, a theoretical upper bound on the Frobenius norm error is given by

$$\min_{\mathbf{Q} \in \mathrm{O}(m)} \left\| \widehat{\mathbf{M}} - \mathbf{Q}\mathbf{M} \right\|_F \leq 2\sqrt{m} \left\| \mathbf{M} \right\|_F \tag{1}$$

Further details and the proof of Equation 1 are provided in Appendix C.

## 3.2 CWY transform and inverse approximation

As observed in [36, 29], computing parameterized orthogonal matrices is computationally costly though it can be sped up with numerical methods. In our case, the composition of multiple House-holder transformations can be cast into high-performance matrix-matrix products through the WY and CWY transforms [21, 25]. The following result from [21] allows us to construct an orthogonal matrix by accumulating householder transformations:

**Theorem 1** *Let the matrix $\mathbf{U} \in \mathbb{R}^{m \times r}$ have linearly independent columns. Partition $\mathbf{U}$ by columns as $\mathbf{U} = (\mathbf{u}_1 \mid \mathbf{u}_2 \mid \ldots \mid \mathbf{u}_r)$ and consider the vector $\boldsymbol{\tau} = (\tau_1, \tau_2, \ldots, \tau_r)^\top$ with $\tau_i \neq 0, 1 \leq i \leq r$. Then, there exists a unique nonsingular upper triangular matrix $S \in \mathbb{R}^{r \times r}$ such that*

$$\mathbf{Q_U} = \left( \mathbf{I} - \frac{\mathbf{u}_1 \mathbf{u}_1^\top}{\tau_1} \right) \left( \mathbf{I} - \frac{\mathbf{u}_2 \mathbf{u}_2^\top}{\tau_2} \right) \cdots \left( \mathbf{I} - \frac{\mathbf{u}_r \mathbf{u}_r^\top}{\tau_r} \right) = \mathbf{I} - \mathbf{U}\mathbf{S}^{-1}\mathbf{U}^\top \tag{2}$$

*where $\mathbf{Q_U} \in \mathrm{O}(m)$. $\mathbf{S}$ can be computed following two steps:*

1. $\mathbf{S} :=$ *the upper triangular part of $\mathbf{U}^\top \mathbf{U}$.*

2. *Divide the diagonal elements of $\mathbf{S}$ by two.*

As in the case of many orthogonal parameterization methods [14], there is a matrix inverse to be computed. This fact makes orthogonal parameterization methods non-scalable, since the inverse computation during training and the gradient update computation are resource-intensive. However, in the case of the CWY transform, the inverse can be approximated with a high degree of precision. In order to efficiently compute $\mathbf{S}^{-1}$, Neuman Series are required [14]. We can separate $\mathbf{S} = \mathbf{D} + \mathbf{A} = \mathbf{D}(\mathbf{I} + \mathbf{D}^{-1}\mathbf{A})$ where $\mathbf{D}$ is a diagonal matrix and $\mathbf{A}$ is a strictly upper triangular matrix. The inverse will be:

$$\mathbf{S}^{-1} = \left( \mathbf{I} + \mathbf{D}^{-1}\mathbf{A} \right)^{-1} \mathbf{D}^{-1} = \left( \sum_{i=0}^{\infty} \left( -\mathbf{D}^{-1}\mathbf{A} \right)^i \right) \mathbf{D}^{-1} \approx \mathbf{D}^{-1} - \mathbf{D}^{-1}\mathbf{A}\mathbf{D}^{-1} \tag{3}$$

It can be demonstrated that, since $\mathbf{A} \in \mathbb{R}^{r \times r}$ is strictly upper triangular, then the spectral radius $\rho\left( \mathbf{D}^{-1}\mathbf{A} \right)$ is less than one and we can ensure that the series from Equation 3 always converges. In fact, $\sum_{i=0}^{\infty}(-\mathbf{D}^{-1}\mathbf{A})^i = \sum_{i=0}^{r-1}(-\mathbf{D}^{-1}\mathbf{A})^i$, and the inverse approximation error grows with the number of columns $r$.

Taking the first and second term of the se-ries in order to approximate the inverse only require diagonal inverses, which are very fast to compute. Rearranging Equation 2, the final equa-tion to approximately compute the accumulated householder product is:

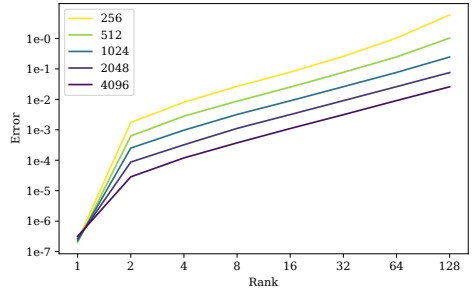

Figure 1: Inverse approximation error

$$\mathbf{Q_U} = \mathbf{I} - \mathbf{U}\mathbf{S}^{-1}\mathbf{U}^\top \approx \mathbf{I} + \mathbf{U}\left(\mathbf{D}^{-1}\mathbf{A}\mathbf{D}^{-1} - \mathbf{D}^{-1}\right)\mathbf{U}^\top \tag{4}$$

To empirically assess the error magnitude, we conducted an experiment approximating a random gaussian accumulated Householder transformation. Figure 1 illustrates how the inverse approximation error varies depending on the rank $r$. The error is defined as $\left\|\mathbf{I} - \mathbf{Q_U}\mathbf{Q_U}^\top\right\|_F / \sqrt{n}$, where $n$ denotes the matrix dimension. As expected, the error is zero when $r = 1$, since the approximation is exact in that case. Although the error grows when increasing $r$, the growth rate remains modest. In particular, for $r \ll m$, the approximation remains remarkably accurate. Further details can be found in Appendix B.

## 3.3 Householder Orthogonal Fine-tuning

Given householder vectors stored in the columns of $\mathbf{U} \in \mathbb{R}^{m \times r}$ and $\mathbf{V} \in \mathbb{R}^{n \times r}$, we construct orthogonal matrices $\mathbf{Q_U} \in \mathrm{O}(m)$ and $\mathbf{Q_V} \in \mathrm{O}(n)$ by applying the CWY transform along with the inverse approximation of $\mathbf{S}$ from Section 3.2. As discussed in Section 3.1, the resulting matrix $\widehat{\mathbf{M}} = \mathbf{Q_U}\mathbf{M}\mathbf{Q_V}$ can express any matrix $\widehat{\mathbf{M}} \in \mathbb{R}^{m \times n}$ such that the hyperspherical energy remains the same as $\mathbf{M} \in \mathbb{R}^{m \times n}$. We call this novel method Householder Orthogonal Fine-tuning (HOFT). As illustrated in Figure 2, our method adapts both $\mathbf{U}, \mathbf{V}^\top$ while preserving the same hyperspherical energy.

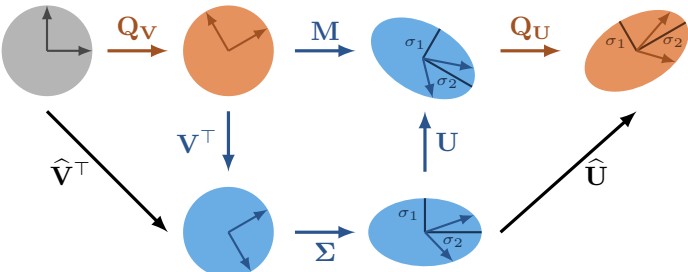

Figure 2: Diagram of our proposed HOFT method

Similar to HRA's rank $r$ [53], HOFT also employs $r$ householder vectors. For both inverse approximations, the computational complexity is $\mathcal{O}\left(r^2(m + n)\right)$, and the matrix-vector multiplications require $\mathcal{O}\left(2mr + 2nr + mn + 2r^2\right)$. Altogether, the total time complexity of HOFT is $\mathcal{O}\left(mn + 2r(m + n) + r^2(m + n + 2)\right) \sim \mathcal{O}\left(mn + (m + n)(r^2 + 2r)\right)$. A comparison of the computational complexity of HOFT to other parameterized orthogonal-based methods is provided in Table 1.

Table 1: Comparisons of parameterized orthogonal-based methods

| Method | #Parameters | Complexity | Coverage |
|--------|-------------|------------|----------|
| OFT | $\frac{m(b-1)}{2}$ | $\mathcal{O}(mn + m(b^2 + b))$ | $b = m$ |
| BOFT | $\frac{mk(b-1)}{2}$ | $\mathcal{O}(mn + mk(b^2 + m))$ | $k = \log m$ and $b = 2$ |
| HRA | $rm$ | $\mathcal{O}(mn + mr)$ | $r = m - 1$ |
| HOFT | $r(m + n)$ | $\mathcal{O}(mn + (m + n)(r^2 + 2r))$ | $r = \max(m, n) - 1$ |

One drawback of OFT is that it requires $b$ to be large in order to achieve $\mathrm{O}(m)$ coverage [36]. The increase of $b$ cannot be arbitrary because of the cost of inverting $b \times b$ matrices. BOFT, on the other hand, offers better coverage at the expense of higher time complexity [29]. HRA provides even better coverage than the two previous methods; however, its Householder transformations must be applied sequentially, and when $\lambda = \infty$, its runtime matches that of OFT [53]. By contrast, HOFT provides the same coverage as HRA, and because most of its computations can be parallelized, it achieves greater speedup and represents an attractive alternative.

Although LoRA and DoRA can be randomly initialized, OFT and BOFT cannot due to the necessity of preserving orthogonality; Cayley's parameterization [14] needs skew-symmetric matrix $\mathbf{R} = \mathbf{0}$ to ensure that the orthogonal parameterized matrix is $\mathbf{Q} = \mathbf{I}$. In general, orthogonal PEFT methods cannot be randomly initialized. However, HOFT and SHOFT can be randomly initialized by considering consecutive pairs of equal vectors $\mathbf{u}_i$. Since they express the same reflection, we can place Householder vectors in the form $\mathbf{U} = (\mathbf{u}_1 \mid \mathbf{u}_1 \mid \cdots \mid \mathbf{u}_k \mid \mathbf{u}_k)$, which yields the identity matrix:

$$\mathbf{Q_U} = \underbrace{\left(\mathbf{I} - \frac{\mathbf{u}_1\mathbf{u}_1^\top}{\tau_1}\right)\left(\mathbf{I} - \frac{\mathbf{u}_1\mathbf{u}_1^\top}{\tau_1}\right)}_{\mathbf{I}} \cdots \underbrace{\left(\mathbf{I} - \frac{\mathbf{u}_k\mathbf{u}_k^\top}{\tau_k}\right)\left(\mathbf{I} - \frac{\mathbf{u}_k\mathbf{u}_k^\top}{\tau_k}\right)}_{\mathbf{I}} = \mathbf{I} \tag{5}$$

Thus, if $r$ is even, we can generate $k = \frac{r}{2}$ pairs of random vectors. If $r$ is odd, we can generate $k = \lfloor\frac{r}{2}\rfloor$ pairs of random vectors and a zero vector. Vectors $\mathbf{u}_i$ are picked from a high-dimensional gaussian distribution. $\mathbf{V}$ is also initialized following this procedure, making $\mathbf{Q_V} = \mathbf{I}$ at the beginning of the training.

## 3.4 Scaled Householder Orthogonal Fine-tuning

The use of scaling transformations in orthogonal fine-tuning methods has been studied in [36] as a way to improve their performance. Drawing also inspiration from DoRA's weight decomposition analysis [27], we propose a variant of HOFT that employs a scaling transformation: Scaled Householder Orthogonal Fine-tuning (SHOFT). As observed in Section 3.1, placing the scaling transformation near the singular value matrix will be interesting from a SVD perspective. Since scaling is performed between two distance preserving transformations, the effect of $\mathbf{m}$ in the singular values of $\mathbf{M}$ is closely controlled. Thus, SHOFT formulation will be as follows

$$\widehat{\mathbf{M}} = \mathbf{Q_U}\mathbf{m}\mathbf{M}\mathbf{Q_V} = \mathbf{Q_U}\mathbf{m}\mathbf{U}\mathbf{\Sigma}\mathbf{V}^\top\mathbf{Q_V} \tag{6}$$

where $\mathbf{Q_U}, \mathbf{Q_V}$ and $\mathbf{m}$ are formed by trainable parameters. It seems more intuitive to be able to redirect with $\mathbf{Q_V}$, transform with $\mathbf{M}$, then scale with $\mathbf{m}$ and finally redirect with $\mathbf{Q_U}$. SHOFT is more flexible since it is no longer constrained to keep the same hyperspherical energy. All elements of vector $\mathbf{m}$ are initialized to one. As observed in other PEFT methods [27, 36], the increase on the amount of trainable parameters due to adding a magnitude vector $\mathbf{m} \in \mathbb{R}^m$ is marginal.

# 4 Experiments

In order to compare HOFT and SHOFT along with other PEFT methods, four main tasks have been selected: commonsense reasoning, machine translation, subject-driven generation and mathematical reasoning. In these tasks, state-of-the-art PEFT methods are evaluated using different pre-trained models to show robustness along different architectures. In addition, quantized models are also employed for evaluating mathematical reasoning. All hyperparameter settings used in the experiments are provided in Appendix A. Additionally, an empirical comparison of time and memory complexity is given in Appendix D.

## 4.1 Commonsense reasoning

For measuring commonsense reasoning performance, we compare HOFT and SHOFT with DoRA and LoRA across eight standard commonsense reasoning benchmarks: BoolQ [8], PIQA [4], SIQA [43], HellaSwag [54], WinoGrande [42], ARC-e [9], ARC-c [9] and OBQA [31]. Following DoRA [27], the training splits of all eight tasks are merged into a single training set, and then each model is evaluated separately on the original test set of each task. The models employed are LLaMA3.1-8B [16], Qwen2.5-7B [50], Phi4-14B [1], and Qwen2.5-14B [50]. We initialize DoRA [27] and LoRA [19] using PiSSA [30]. We set $r = 16$ for all PEFT methods and train the models for two epochs.

The results of each individual task along with the average task accuracy per model and PEFT method are shown in Table 2, where it can be seen that HOFT and SHOFT generally achieve higher scores than LoRA and DoRA across most models, with SHOFT performing comparably to DoRA for

Qwen2.5-7B. Moreover, both HOFT and SHOFT continue to deliver strong results as model size grows, demonstrating solid performance on both Phi4-14B and Qwen2.5-14B. In particular, HOFT and SHOFT attain the highest scores on nearly every task, matching LoRA and DoRA only on PIQA and ARC-e. This underscores their robustness and efficiency when trained on datasets containing multiple domains.

Table 2: Accuracy comparison (%) on various commonsense reasoning benchmarks

| Model | Method | #Params (%) | BoolQ | PIQA | SIQA | HellaSwag | WinoGrande | ARC-e | ARC-c | OBQA | Avg. |
|---|---|---|---|---|---|---|---|---|---|---|---|
| LLaMA3.1-8B | LoRA | 0.35 | 88.2 | **88.5** | 80.3 | 96.7 | 80.5 | 91.9 | 82.3 | 87.4 | 87.0 |
| | DoRA | 0.36 | 88.1 | 89.1 | 80.1 | 96.6 | **81.4** | 92.0 | 82.5 | 86.8 | 87.1 |
| | HOFT | 0.35 | 88.5 | **88.5** | 80.9 | 96.8 | 80.4 | 92.7 | 83.2 | 88.4 | 87.4 |
| | SHOFT | 0.36 | **88.8** | **88.5** | 80.1 | 96.8 | 81.2 | 92.0 | 82.9 | 86.6 | 87.1 |
| Qwen2.5-7B | LoRA | 0.35 | 88.4 | 89.5 | 79.6 | 96.8 | 82.5 | 95.8 | 88.7 | 92.2 | 89.2 |
| | DoRA | 0.36 | 88.9 | 89.8 | 79.2 | 96.8 | 82.5 | 96.2 | 88.9 | 92.4 | 89.3 |
| | HOFT | 0.35 | 89.0 | 89.1 | 79.2 | 96.4 | 80.4 | 95.9 | 88.4 | 92.4 | 88.9 |
| | SHOFT | 0.36 | 88.8 | 89.5 | 79.5 | 96.5 | 80.7 | 95.7 | 89.1 | 93.4 | 89.2 |
| Phi4-14B | LoRA | 0.33 | 89.7 | 92.0 | 81.7 | 97.3 | **87.9** | 97.9 | 93.1 | 94.2 | 91.7 |
| | DoRA | 0.35 | 90.0 | 91.9 | 82.0 | **97.4** | 87.3 | 98.0 | 93.5 | 94.0 | 91.8 |
| | HOFT | 0.33 | **90.1** | **92.7** | **82.3** | **97.4** | 86.7 | **98.1** | 94.3 | 93.6 | 91.9 |
| | SHOFT | 0.35 | 90.0 | **92.7** | 81.9 | 97.3 | 87.4 | 98.0 | **94.5** | **95.4** | **92.2** |
| Qwen2.5-14B | LoRA | 0.31 | 89.9 | **92.7** | 82.1 | 98.0 | 87.1 | **98.1** | 93.6 | 95.0 | 92.1 |
| | DoRA | 0.32 | 89.9 | 92.5 | 82.6 | 98.0 | 87.3 | **98.1** | 93.0 | 94.6 | 92.0 |
| | HOFT | 0.31 | 90.2 | 91.9 | **83.8** | 98.0 | 87.6 | 97.7 | **93.7** | **96.2** | **92.4** |
| | SHOFT | 0.32 | **90.3** | 92.3 | 83.0 | **98.1** | **88.2** | 97.2 | 92.7 | **96.2** | 92.3 |

## 4.2 Machine Translation

For measuring machine translation performance, HOFT and SHOFT are compared with DoRA and LoRA using four languages from the CoVoST 2 [47] dataset: Slovene, German, Latvian and French. We chose these languages in order to have two well-represented languages and two low-resource languages. For French and German, models are trained on the first 10K elements of the training split. Three models are adapted for this task: NLLB-3.3B [11], LLaMA2-7B [45], and LLaMA3.1-8B [16]. We set $r = 16$ for all PEFT methods and train the models for 2 epochs. Both BLEU [33, 35] and COMET [39, 38] results are provided for each individual language per model and PEFT method. Results obtained are shown in Table 3. We additionally provide baseline performance of the models.

Table 3: Performance comparison on X → English machine translation tasks

| Model | Method | #Params (%) | Slovene | | German | | Latvian | | French | |
|---|---|---|---|---|---|---|---|---|---|---|
| | | | BLEU | COMET | BLEU | COMET | BLEU | COMET | BLEU | COMET |
| NLLB-3.3B | Baseline | - | 39.7 | 87.5 | 39.3 | 86.2 | 31.2 | 81.3 | 38.5 | 84.9 |
| | LoRA | 0.42 | 46.8 | 89.2 | 44.5 | **87.7** | 38.2 | 83.9 | **49.7** | **87.8** |
| | DoRA | 0.43 | 46.8 | 89.1 | **44.7** | 87.6 | 38.2 | 83.9 | 49.5 | 87.7 |
| | HOFT | 0.42 | **48.0** | 89.4 | 44.4 | 87.6 | 38.6 | 83.9 | 49.5 | 87.7 |
| | SHOFT | 0.43 | 46.4 | **89.5** | 44.5 | **87.7** | **38.7** | **84.0** | **49.7** | **87.8** |
| LLaMA2-7B | 0-shot | - | 26.8 | 72.8 | 30.4 | 74.1 | 4.5 | 52.2 | 37.2 | 79.3 |
| | LoRA | 0.19 | 39.3 | 84.7 | 41.5 | 86.9 | 15.5 | 66.2 | 47.0 | 87.2 |
| | DoRA | 0.19 | 39.6 | 84.8 | 41.4 | 86.9 | **16.2** | **66.6** | 47.0 | 87.2 |
| | HOFT | 0.19 | 40.6 | 85.2 | 41.4 | 86.9 | 15.8 | **66.6** | 47.0 | **87.3** |
| | SHOFT | 0.19 | **41.2** | **85.6** | **41.6** | **87.0** | 15.7 | 65.9 | **47.1** | **87.3** |
| LLaMA3.1-8B | 0-shot | - | 34.2 | 77.8 | 40.9 | 86.2 | 22.9 | 70.8 | 41.6 | 82.7 |
| | LoRA | 0.12 | 36.2 | 84.1 | 42.3 | 87.4 | 32.7 | **80.9** | **46.8** | 85.5 |
| | DoRA | 0.12 | 42.4 | 85.0 | 42.2 | 87.4 | **32.8** | 80.8 | 46.7 | 85.5 |
| | HOFT | 0.12 | **44.2** | **86.6** | 42.9 | 87.5 | 32.2 | 80.4 | 46.7 | **85.6** |
| | SHOFT | 0.12 | 43.6 | 86.4 | **43.1** | **87.7** | 31.9 | 80.4 | **46.8** | **85.6** |

From Table 3 we can observe how HOFT and SHOFT provide competitive results in French and German. In Latvian, HOFT and SHOFT give similar results in the case of NLLB-3.3B. For Slovene, both methods clearly outperform LoRA and DoRA with LLaMA2-7B, while HOFT in BLEU and SHOFT in COMET with NLLB-3.3B. Notably, the difference on both metrics is significantly higher

with LLaMA3.1-8B. Overall, the top BLEU and COMET scores are almost always achieved by either HOFT or SHOFT, underlining their effectiveness across multiple languages.

## 4.3 Subject-driven generation

For subject-driven generation, we follow the experimental protocol of HRA [53], using the Dream-Booth dataset [41] to train and evaluate on 25 distinct subjects, each with 30 associated prompts. We adapt the pre-trained Stable Diffusion (SD) model [40] and compare PEFT methods quantitatively across four metrics: subject fidelity (DINO [7] and CLIP-I [37]), prompt fidelity (CLIP-T [37]), and sample diversity (LPIPS [56]).

Table 4: Quantitative comparison of subject-driven generation

| Method | #Param (M) | DINO $\uparrow$ | CLIP-I $\uparrow$ | CLIP-T $\uparrow$ | LPIPS $\uparrow$ |
|---|---|---|---|---|---|
| Real Images | – | 0.764 | 0.890 | – | 0.562 |
| DreamBooth | 859.52 | 0.614 | 0.778 | 0.239 | 0.737 |
| LoRA | 0.80 | 0.613 | 0.765 | 0.237 | 0.744 |
| COFT$_{b=4}$ | 23.3 | 0.630 | 0.783 | 0.235 | 0.744 |
| OFT$_{b=4}$ | 23.3 | 0.632 | 0.785 | 0.237 | 0.746 |
| HRA$_{r=7,8,\lambda=0}$ | 0.69 | 0.670 | 0.803 | 0.238 | 0.758 |
| HRA$_{r=7,8,\lambda=10^{-3}}$ | 0.69 | 0.661 | 0.799 | 0.255 | 0.760 |
| HRA$_{r=7,8,\lambda=\infty}$ | 0.69 | 0.651 | 0.794 | **0.274** | **0.778** |
| HOFT$_{r=2}$ | 0.40 | 0.657 | 0.793 | 0.239 | 0.758 |
| SHOFT$_{r=2}$ | 0.41 | 0.658 | 0.793 | 0.241 | 0.757 |
| HOFT$_{r=4}$ | 0.80 | **0.680** | **0.810** | 0.235 | 0.752 |
| SHOFT$_{r=4}$ | 0.81 | **0.680** | 0.808 | 0.235 | 0.747 |

The results, together with the provided baselines, are summarized in Table 4. Both HOFT and SHOFT outperform all baselines in subject fidelity. In terms of textual prompt fidelity, they achieve results comparable with LoRA, OFT, and COFT. For sample diversity, they also deliver competitive performance. Additionally, we also tested HOFT and SHOFT at half the rank. Even with fewer trainable parameters, both methods consistently outperform LoRA, OFT, and COFT across all metrics, while remaining competitive with HRA on subject fidelity.

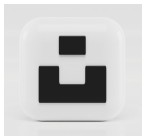 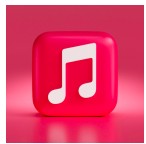 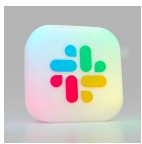   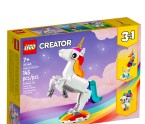 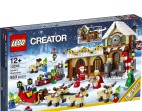 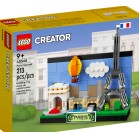

images of 3d icons                                        images of lego sets

Figure 3: Examples of training images of 3D icons and lego sets

Therefore, in order to gain a deeper insight into subject fidelity, we conducted an additional experiment following DoRA [27]. We fine-tuned a pre-trained Stable Diffusion XL (SDXL) model [34] on two datasets: 3D icons and lego sets. In Figure 3 we can see some examples of the styles to be learned. In this experiment, five PEFT methods are used for evaluation: LoRA, HRA, OFT, HOFT, and SHOFT. To ensure a fair comparison, all methods used the same random sample seed for generating the images.

As shown in Figure 4, HOFT and SHOFT provide better personalization than LoRA, HRA, and OFT. When generating 3D icons, both methods closely match the style and subject of the training images. This highlights the value of orthogonality: while OFT also produces competitive results, LoRA and HRA struggle to generate realistic 3D icons. Moreover, HOFT and SHOFT produce accurate text in the lego sets, while the rest do not achieve it. Additional qualitative examples can be found in Appendix E.

Prompt: a TOK 3d icon of an orange llama eating ramen, in the style of TOK

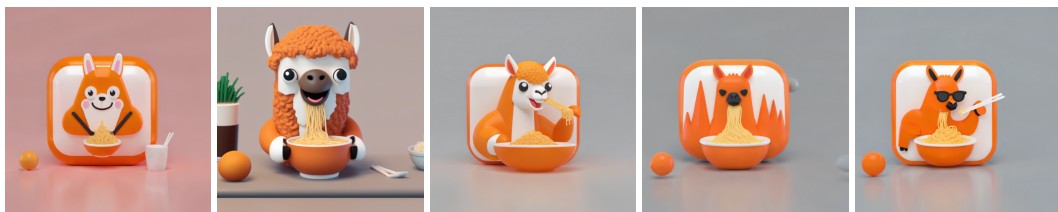

Prompt: a TOK lego set of an orange llama eating ramen, in the style of TOK

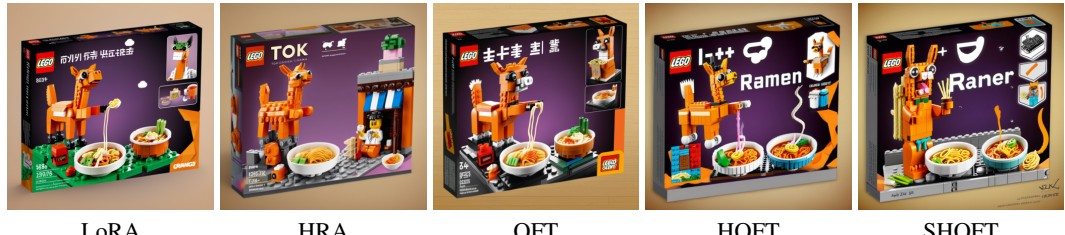

| LoRA | HRA | OFT | HOFT | SHOFT |

Figure 4: Qualitative results on lego sets and 3d icons datasets

## 4.4 Mathematical reasoning

For the mathematical reasoning experiments, we follow the HRA guidelines [53]. We fine-tune LLaMA2-7B [45] on the MetaMathQA dataset [52], which contains a diverse amount of mathematical questions along with rationalized answers. HOFT and SHOFT are evaluated on the GSM8K [10] and MATH [52] validation sets. Table 5 shows the accuracy of these methods alongside other PEFT baselines.

Table 5: Accuracy comparison (%) on mathematical reasoning datasets

| Method | GSM8K | MATH |
|---|---|---|
| Baseline | 14.6 | 2.5 |
| LoRA | 50.2 | 7.8 |
| OFT | 50.1 | 8.4 |
| BOFT | 50.6 | 8.6 |
| PiSSA | 53.1 | 7.4 |
| HRA | 56.3 | 9.3 |
| HOFT | **56.6** | 8.9 |
| SHOFT | 55.0 | **9.8** |

The results in Table 5 show that HOFT and SHOFT are competitive with existing PEFT methods on mathematical reasoning benchmarks. HOFT achieves the highest accuracy on GSM8K, while SHOFT achieves the best score on the more challenging MATH dataset. This suggests that the scaling transformation plays a role to improve performance on harder math questions.

## 4.5 QHOFT: Quantized HOFT

In addition to the previous mathematical reasoning experiment, two additional experiments are performed in order to test the quantized versions of HOFT and SHOFT. We adapt 4-bit quantized [12] LLaMA2-7B [45] and LLaMA3.1-8B [16] to GSM8K [10] and Orca-Math [32] datasets separately and evaluate them on their respective test datasets. In particular, we follow DoRA [27] Orca-Math experimental setup: 100K elements for training and 2K for evaluation. The experimental results are reported in Table 6.

Table 6: Accuracy comparison (%) on mathematical reasoning datasets using quantized models

| Model | Method | #Params (%) | GSM8K | Orca-Math |
|---|---|---|---|---|
| LLaMA2-7B | QLoRA | 0.19 | 27.9 | 14.4 |
| | QDoRA | 0.19 | 29.0 | 13.0 |
| | QHOFT | 0.19 | **30.5** | 14.7 |
| | QSHOFT | 0.19 | 29.3 | **15.5** |
| LLaMA3.1-8B | QLoRA | 0.12 | 53.8 | 54.1 |
| | QDoRA | 0.12 | 56.5 | 53.8 |
| | QHOFT | 0.12 | 55.0 | **57.2** |
| | QSHOFT | 0.12 | **57.0** | 54.6 |

The results in Table 6 demonstrate that the quantized versions of HOFT and SHOFT consistently outperform QLoRA and QDoRA under extreme parameter efficiency. On LLaMA2-7B, QHOFT achieves the highest GSM8K accuracy, while QSHOFT leads on Orca-Math. On the larger LLaMA3.1-8B model, QSHOFT delivers the best GSM8K performance, and QHOFT achieves the best Orca-Math score. These results confirm that QHOFT and QSHOFT perform well even with aggressive 4-bit quantization.

# 5 Limitations

One limitation of our work is the challenge of adapting architectures with low-dimensional weight matrices: neither HOFT nor SHOFT can fully enforce orthogonality in their learned weights when the dimensionality is low. Although both methods achieve a slightly lower peak memory usage than DoRA, their memory footprint remains substantially higher than that of LoRA.

# 6 Conclusions

In this work, we examined some of the theoretical foundations of orthogonal fine-tuning. Based on our findings we proposed HOFT, a new PEFT method that adapts a pre-trained weight matrix by pre- and post-multiplying it with learned orthogonal matrices. We also developed SHOFT, a HOFT variant that introduces scaling transformations to further improve performance. Both exhibit good theoretical properties and provide higher flexibility. Our experimental results show that HOFT and SHOFT consistently match or outperform leading PEFT approaches across a wide range of benchmarks. To the best of our knowledge, QHOFT and QSHOFT are the first quantized orthogonal fine-tuning methods that maintain the benefits of their non-quantized counterparts, while operating with substantially reduced time and memory requirements.

For future work, we would like to extend our evaluation to include visual instruction tuning and the adaptation of multi-modal pre-trained models. In addition, we plan to explore how to reduce the number of trainable parameters in both methods, for instance by adopting vector-bank strategies similar to VB-LoRA. Finally, as discussed in Section 5, we would like to develop a variant of HOFT optimized for smaller weight matrices, aiming to reduce memory overhead and enforce orthogonality constraints.

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

# A Experimental details

## A.1 Commonsense reasoning experiments

For commonsense reasoning experiments, we employ a NVIDIA A40 GPU for training LLaMA3.1-8B and Qwen2.5-7B models. For training Phi4-14B and Qwen2.5-14B, a NVIDIA H100 GPU was employed. For all experiments, the rank $r$ was set to 16, and a dropout of 0.05. The optimizer employed was AdamW with Linear LR Scheduler. All models were trained for 2 epochs using a batch size of 4 and accumulation step of 4. The number of warmup steps was set to 100. The adapted layers were Query, Key, Value, Up, Down and Gate. We provide in Table 7 the learning rates used per model and per PEFT method.

Table 7: Learning rate hyperparameter configurations for commonsense reasoning experiments

| Method | LLaMA3.1-8B | Qwen2.5-7B | Phi4-14B | Qwen2.5-14B |
|--------|-------------|------------|----------|-------------|
| LoRA   | 9e-5 | 1e-4 | 9e-5 | 1e-4 |
| DoRA   | 1e-4 | 9e-5 | 9e-5 | 9e-5 |
| HOFT   | 1e-4 | 9e-5 | 9e-5 | 9e-5 |
| SHOFT  | 2e-4 | 1e-4 | 9e-5 | 2e-4 |

## A.2 Machine translation experiments

For machine translation experiments, we use a NVIDIA A30 GPU for training NLLB-3.3B model. For training LLaMA2-7B and LLaMA3.1-8B, a NVIDIA A40 GPU was used. For all experiments, the rank $r$ was set to 16, and a dropout of 0.05. The optimizer employed was AdamW with Linear LR Scheduler. For French and German datasets, models were trained for 2 epochs on the first 10K elements of the training dataset. For Slovene and Latvian, models were trained for 3 epochs. All experiments use batch size of 16 and accumulation step of 4. The number of warmup steps was set to 100. The adapted layers were Query, Key and Value. We provide in Table 8 the learning rates used per language, model and per PEFT method.

Table 8: Learning rate hyperparameter configurations for machine translation experiments

| Language | Method | NLLB-3.3B | LLaMA2-7B | LLaMA3.1-8B |
|----------|--------|-----------|-----------|-------------|
| Slovene | LoRA  | 4e-4 | 4e-4 | 8e-4 |
|         | DoRA  | 4e-4 | 4e-4 | 9e-4 |
|         | HOFT  | 5e-4 | 6e-4 | 1e-3 |
|         | SHOFT | 5e-4 | 6e-4 | 7e-4 |
| German  | LoRA  | 6e-4 | 3e-4 | 4e-4 |
|         | DoRA  | 6e-4 | 3e-4 | 4e-4 |
|         | HOFT  | 2e-4 | 2e-4 | 8e-4 |
|         | SHOFT | 2e-4 | 2e-4 | 4e-4 |
| French  | LoRA  | 5e-4 | 1e-4 | 1e-4 |
|         | DoRA  | 4e-4 | 1e-4 | 1e-4 |
|         | HOFT  | 1e-4 | 1e-4 | 3e-4 |
|         | SHOFT | 4e-4 | 1e-4 | 1e-4 |
| Latvian | LoRA  | 5e-4 | 4e-4 | 5e-4 |
|         | DoRA  | 5e-4 | 5e-4 | 5e-4 |
|         | HOFT  | 2e-4 | 9e-4 | 6e-4 |
|         | SHOFT | 3e-4 | 8e-4 | 5e-4 |

### A.3 Subject-driven generation experiments

For quantitative subject-driven experiments, we employ 10 NVIDIA A40 GPUs for training the Stable Diffusion 1.5 model. For all experiments, no dropout was used. The optimizer employed was AdamW with Linear LR Scheduler. All models were trained for 2005 steps using a batch size of 1. The adapted layers were Query, Key, Value and Out from the U-Net part. The learning rate used for training both HOFT and SHOFT is 5e-4.

For qualitative subject-driven experiments, we employ 5 NVIDIA A40 GPUs for training the Stable Diffusion XL model. For all experiments, no dropout was used. For all PEFT methods the rank $r$ was set to 16, except for HRA, which was set to 32 for fair comparison. The optimizer employed was AdamW with Linear LR Scheduler. All models were trained 1000 steps using a batch size of 4 and gradient accumulation of 4. The adapted layers were Query, Key, Value and Out from the U-Net and text encoder part. The learning rate used for training both all PEFT methods is 1e-4.

### A.4 Mathematical reasoning experiments

For mathematical reasoning experiments, we employ a NVIDIA H100 GPU for training LLaMA2-7B model. For all experiments, the rank $r$ was set to 8, and no dropout. The optimizer employed was AdamW with Linear LR Scheduler. All models were trained for 2 epochs using a batch size of 8 and accumulation step of 2. The warmup ratio was set to 0.05. The adapted layers were Query and Value. The learning rates used by HOFT and SHOFT were 1e-3 and 7e-4 respectively.

### A.5 Mathematical reasoning experiments with quantized models

For experiments in mathematical reasoning with quantized models, we employ a NVIDIA H100 GPU for training LLaMA2-7B and LLaMA3.1-8B models. Models are quantized using NF4 and double quatization. For all experiments, the rank $r$ was set to 16, and a dropout of 0.05. The optimizer employed was AdamW with Linear LR Scheduler. For Orca-Math dataset, all models were trained for 2 epochs using a batch size of 4 and accumulation step of 1. For GSM8K dataset, all models were trained for 3 epochs using a batch size of 4 and accumulation step of 1. The number of warmup steps was set to 100. The adapted layers were Query, Key and Value. We provide in Table 9 the learning rates used per model and per PEFT method.

Table 9: Learning rate hyperparameter configurations for mathematical reasoning on quantized models

| Dataset | Method | GSM8K | Orca-Math |
|---------|--------|-------|-----------|
| LLaMA2-7B | QLoRA | 4e-4 | 1e-4 |
| | QDoRA | 4e-4 | 9e-5 |
| | QHOFT | 3e-4 | 1e-4 |
| | QSHOFT | 3e-4 | 4e-4 |
| LLaMA3.1-8B | QLoRA | 2e-4 | 9e-5 |
| | QDoRA | 1e-4 | 9e-5 |
| | QHOFT | 3e-4 | 2e-4 |
| | QSHOFT | 4e-4 | 2e-4 |

# B  About inverse approximation error

## B.1  Hyperspherical energy difference

Given $\mathbf{W} = (\mathbf{w}_1 \mid \cdots \mid \mathbf{w}_n) \in \mathbb{R}^{m \times n}$, where $\mathbf{w}_i$ denotes the $i$-th column of matrix $\mathbf{W}$, the hyperspherical energy is defined as follows:

$$\mathrm{HE}(\mathbf{W}) = \sum_{i \neq j} \| \mathbf{w}_i - \mathbf{w}_j \|^{-1} \tag{7}$$

In order to measure the difference on the hyperspherical energy, we conduct an experiment by approximating two random gaussian accumulated householder transformations $\mathbf{Q_U}, \mathbf{Q_V}$. We measure $|\mathrm{HE}(\mathbf{M}) - \mathrm{HE}(\mathbf{Q_U M Q_V})|$, where $\mathbf{M}$ is a random gaussian matrix. Results are show in Figure 5.

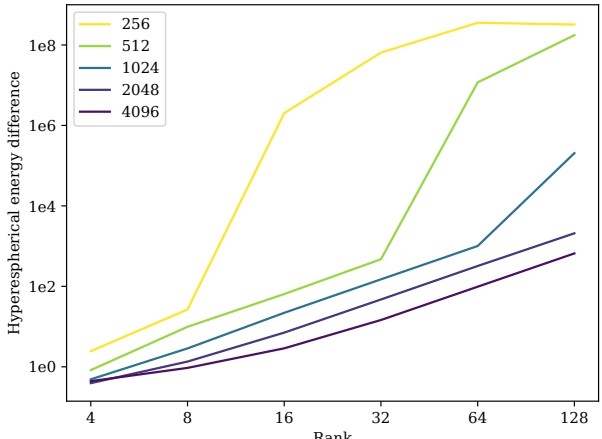

Figure 5: Hyperspherical energy difference

As observed in Figure 5, the hyperspherical energy tends to increase rapidly for higher ranks. Remarkably, for all cases the difference is negligible when $r = 1$ and $r = 2$ (omitted in Figure 5 for clarity). We can conclude from Figures 1 and 5 that, for a given rank $r$, the inverse approximation improves as the dimension of the matrix increases. Given the growing tendency for weight matrices in new pre-trained models, this is really convenient.

## B.2  Indifference towards weight decay

One theoretical property of computing the CWY transform along with the inverse approximation is that, after applying weight decay to the original weights $\mathbf{U} \in \mathbb{R}^{m \times r}$, the resulting accumulated householder matrix remains the same. That is, given $\mathbf{U}' = \mathbf{U} - \lambda \mathbf{U}$, we compute

$$
\begin{aligned}
\mathbf{Q_{U'}} &= \mathbf{I} + \mathbf{U}' \left( \mathbf{D}'^{-1} \mathbf{A}' \mathbf{D}'^{-1} - \mathbf{D}'^{-1} \right) \mathbf{U}'^{\top} \\
&= \mathbf{I} + (1 - \lambda)^2 \mathbf{U} \left( \frac{1}{(1-\lambda)^2} \mathbf{D}^{-1} (1-\lambda)^2 \mathbf{A} \frac{1}{(1-\lambda)^2} \mathbf{D}^{-1} - \frac{1}{(1-\lambda)^2} \mathbf{D}^{-1} \right) \mathbf{U}^{\top} \\
&= \mathbf{I} + \frac{(1-\lambda)^2}{(1-\lambda)^2} \mathbf{U} \left( \mathbf{D}^{-1} \mathbf{A} \mathbf{D}^{-1} - \mathbf{D}^{-1} \right) \mathbf{U}^{\top} \\
&= \mathbf{I} + \mathbf{U} \left( \mathbf{D}^{-1} \mathbf{A} \mathbf{D}^{-1} - \mathbf{D}^{-1} \right) \mathbf{U}^{\top} = \mathbf{Q_U}
\end{aligned}
$$

Thus, we ensure that distance-preserving transformations in HOFT and SHOFT are not affected by weight decay. From this fact, we can ignore weight decay when adapting with HOFT. Additionally, when adapting with SHOFT, weight decay only affects the scaling transformation $\mathbf{m}$.

## C  Proof for Equation 1

Given $\mathbf{M}, \widehat{\mathbf{M}} \in \mathbb{R}^{m \times n}$ two matrices such that both have the same hyperspherical energy. Then

$$\min_{\mathbf{Q} \in O(m)} \left\| \widehat{\mathbf{M}} - \mathbf{Q}\mathbf{M} \right\|_F = \min_{\mathbf{Q} \in O(m)} \left\| \widehat{\mathbf{Q}}_{\mathbf{U}} \mathbf{M} \widehat{\mathbf{Q}}_{\mathbf{V}} - \mathbf{Q}\mathbf{M} \right\|_F \leq \left\| \mathbf{M} \widehat{\mathbf{Q}}_{\mathbf{V}} - \mathbf{M} \right\|_F$$

$$= \left\| \mathbf{M} \left( \widehat{\mathbf{Q}}_{\mathbf{V}} - \mathbf{I} \right) \right\|_F \leq \|\mathbf{M}\|_F \left\| \widehat{\mathbf{Q}}_{\mathbf{V}} - \mathbf{I} \right\|_F$$

Now we need to compute $\left\| \widehat{\mathbf{Q}}_{\mathbf{V}} - \mathbf{I} \right\|_F$

$$\left\| \widehat{\mathbf{Q}}_{\mathbf{V}} - \mathbf{I} \right\|_F^2 = \mathrm{Tr} \left( \left( \widehat{\mathbf{Q}}_{\mathbf{V}} - \mathbf{I} \right)^\top \left( \widehat{\mathbf{Q}}_{\mathbf{V}} - \mathbf{I} \right) \right) = \mathrm{Tr} \left( \widehat{\mathbf{Q}}_{\mathbf{V}}^\top \widehat{\mathbf{Q}}_{\mathbf{V}} - \widehat{\mathbf{Q}}_{\mathbf{V}}^\top - \widehat{\mathbf{Q}}_{\mathbf{V}} + \mathbf{I} \right) =$$

$$= 2m - \mathrm{Tr} \left( \widehat{\mathbf{Q}}_{\mathbf{V}}^\top \right) - \mathrm{Tr} \left( \widehat{\mathbf{Q}}_{\mathbf{V}} \right) = 2m - 2\mathrm{Tr} \left( \widehat{\mathbf{Q}}_{\mathbf{V}} \right)$$

The previous expression attains its maximum precisely when $\widehat{\mathbf{Q}}_{\mathbf{V}} = -\mathbf{I}$. In that case, we conclude that $\mathrm{Tr} \left( \widehat{\mathbf{Q}}_{\mathbf{V}} \right) = -m$ and consequently $\left\| \widehat{\mathbf{Q}}_{\mathbf{V}} - \mathbf{I} \right\|_F^2 \leq 4m$. Thus, final upper-bound will be

$$\min_{\mathbf{Q} \in O(m)} \left\| \widehat{\mathbf{M}} - \mathbf{Q}\mathbf{M} \right\|_F \leq 2\sqrt{m} \|\mathbf{M}\|_F$$

## D  Time and memory consumption

In order to give a better understanding of the time and memory complexity of HOFT and SHOFT, we provide the runtime for training and the peak memory usage during training from the commonsense reasoning, qualitative subject-driven generation and mathematical reasoning using quantized models experiments. All values are gathered in Tables 10, 11 and 12.

In Table 10 we can observe that both HOFT and SHOFT are 72.5% and 55.3% faster on average than DoRA, respectively. With respect to LoRA, they are on average 35.1% and 41.8% slower, respectively. In terms of memory, both HOFT and SHOFT peak memories are between LoRA's and DoRA's peak memories, except in Phi4-14B, where the memory is higher in HOFT and SHOFT. This unusual peak in Phi4-14B is due to the fact Query, Key and Value are all together in a matrix (the same happens with Up and Gate layers).

Table 10: Memory and time complexity comparison on the commonsense reasoning task

| Model | Method | Training time (hours) | Peak memory (GB) |
|---|---|---|---|
| LLaMA3.1-8B | LoRA | 1.5 | 31.9 |
| | DoRA | 3.3 | 45.8 |
| | HOFT | 2.3 | 42.3 |
| | SHOFT | 2.6 | 44.5 |
| Qwen2.5-7B | LoRA | 6.1 | 30.7 |
| | DoRA | 13.9 | 44.5 |
| | HOFT | 8.0 | 41.3 |
| | SHOFT | 9.2 | 43.5 |
| Phi4-14B | LoRA | 2.4 | 49.9 |
| | DoRA | 8.3 | 68.0 |
| | HOFT | 3.4 | 78.6 |
| | SHOFT | 3.7 | 78.0 |
| Qwen2.5-14B | LoRA | 2.8 | 39.8 |
| | DoRA | 7.6 | 59.4 |
| | HOFT | 5.9 | 56.4 |
| | SHOFT | 6.4 | 57.6 |

528

From Table 11, both HOFT and SHOFT are 72.1% faster than OFT, and 732.5% faster than HRA. In this respect, it is worth noting that HRA entails a number of sequential householder transformations that leads to a comparatively high training time. With respect to LoRA, they are on average 32.6% slower. In terms of memory, both HOFT and SHOFT require less memory than OFT and HRA, and the same as LoRA.

Table 11: Memory and time complexity comparison on the mathematical reasoning using quantized models experiments

| Method | Training time (hours) | Peak memory (GB) |
|---|---|---|
| LoRA | 2.9 | 25.5 |
| HRA | 35.8 | 25.7 |
| OFT | 7.4 | 26.7 |
| HOFT | 4.3 | 25.5 |
| SHOFT | 4.3 | 25.5 |

Table 12 shows that there is a minor difference in time cost between LoRA, HOFT and SHOFT. In the case of DoRA, it is 16.7% slower than the rest. In terms of memory, both HOFT and SHOFT peak memories are between LoRA's and DoRA's peak memories, requiring at most 9.6% and 20.8% more memory than LoRA, respectively.

Table 12: Memory and time complexity comparison on the mathematical reasoning using quantized models experiments

| Model | Method | Training time (hours) | Peak memory (GB) |
|---|---|---|---|
| LLaMA2-7B | QLoRA | 0.9 | 43.2 |
| | QDoRA | 1.2 | 58.2 |
| | QHOFT | 1.0 | 47.7 |
| | QSHOFT | 1.0 | 52.2 |
| LLaMA3.1-8B | QLoRA | 1.0 | 52.0 |
| | QDoRA | 1.2 | 60.4 |
| | QHOFT | 1.0 | 57.0 |
| | QSHOFT | 1.0 | 56.4 |

# E Additional experiments

## E.1 Rank exploration

We explore the effect of various rank settings $r \in \{2, 4, 8, 16, 32, 64\}$ on LoRA, DoRA, HOFT and SHOFT by evaluating the fine-tuned LLaMA3.1-8B and Qwen2.5-7B performance on the commonsense reasoning tasks described in Section 4.1. The implementation settings are the same as those for rank 16, given in Appendix A.

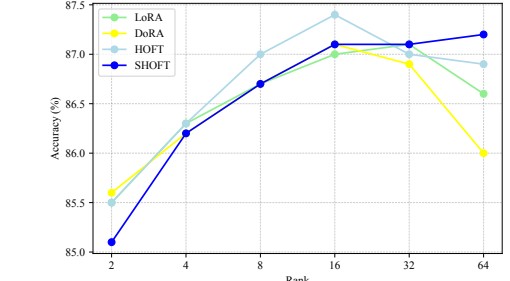 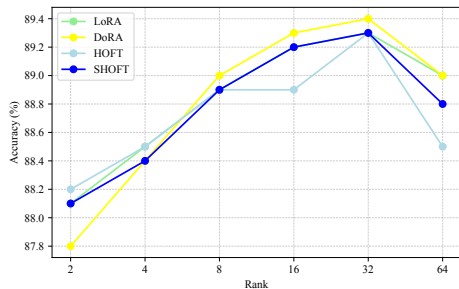

Figure 6: Rank exploration in LLaMA3.1-8B    Figure 7: Rank exploration in Qwen2.5-7B

The average accuracies of the PEFT methods across different ranks are shown in Figures 6 and 7. In Figure 6, all four methods improve sharply up to $r = 16$, where HOFT peaks. Beyond $r = 16$, DoRA's performance declines markedly while LoRA falls slightly. In contrast, SHOFT maintains a mild upward trend approaching HOFT best result for $r = 64$. In Figure 7, the methods again climb to a peak at $r = 32$, where DoRA attains the highest accuracy, with SHOFT and LoRA close behind. At $r = 64$, HOFT's accuracy falls more noticeably, whereas the others dip only slightly.

Overall, these results suggest that HOFT is the strongest option for moderate ranks, but SHOFT is the most robust method at higher ranks and offers the steadiest, most consistent gains across the entire rank spectrum.

## E.2 More qualitative results on subject-driven generation

Prompt: a TOK pink 3d icon of a rainbow unicorn eating marshmallow, in the style of TOK

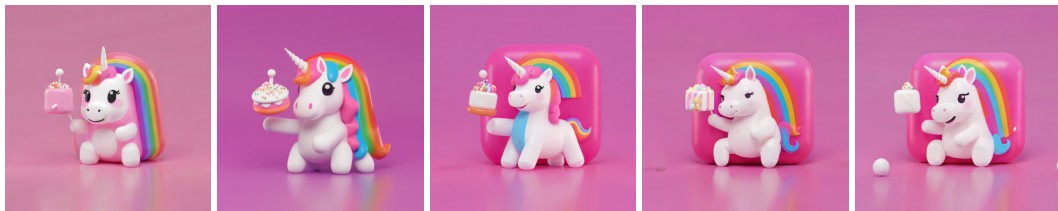

Prompt: a TOK 3d icon of a yellow racoon eating banana, in the style of TOK

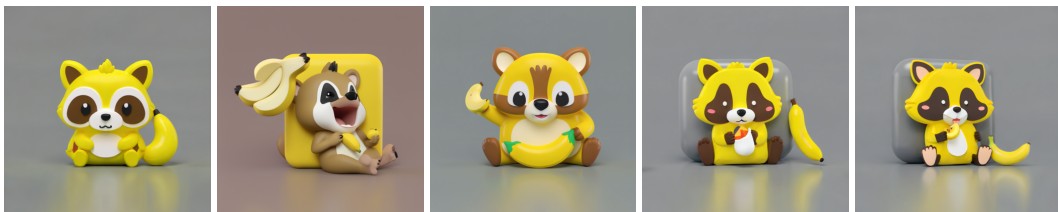

Prompt: a TOK 3d icon of a yellow duck eating sushi, in the style of TOK

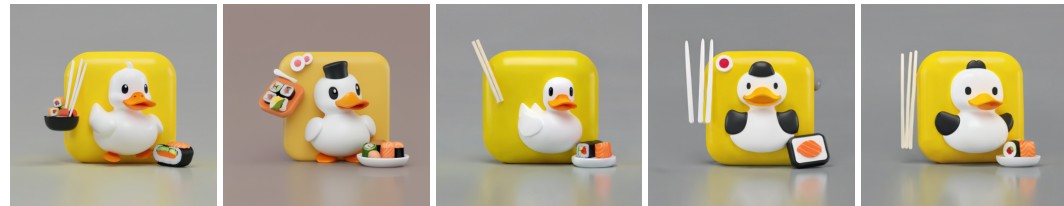

Prompt: a TOK 3d icon of a demon red panda eating bamboo, in the style of TOK

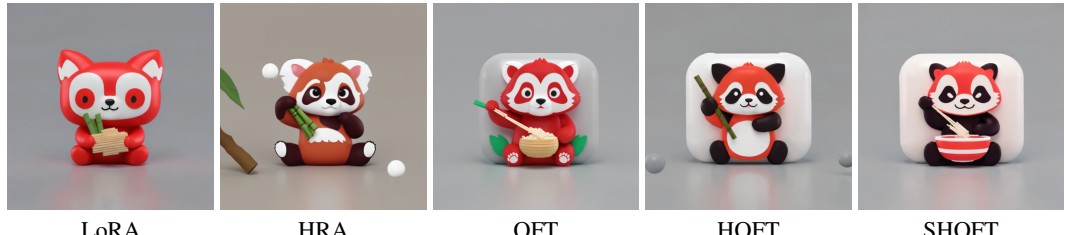

| LoRA | HRA | OFT | HOFT | SHOFT |

Figure 8: Comparison of different prompts in 3D icons dataset

Prompt: a TOK lego set of a colorful coral reef with an explorer submarine and a giant octopus, in the style of TOK

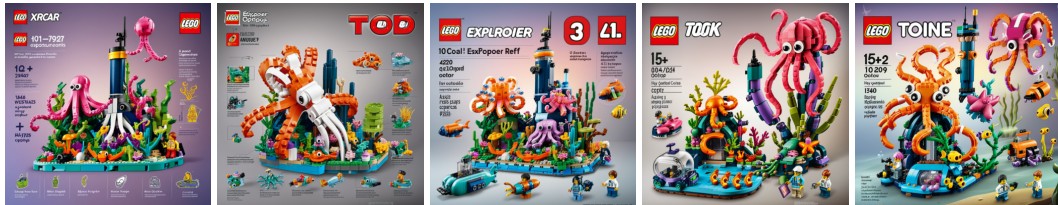

Prompt: a TOK lego set of a crashed spaceship turned into a jungle village, in the style of TOK

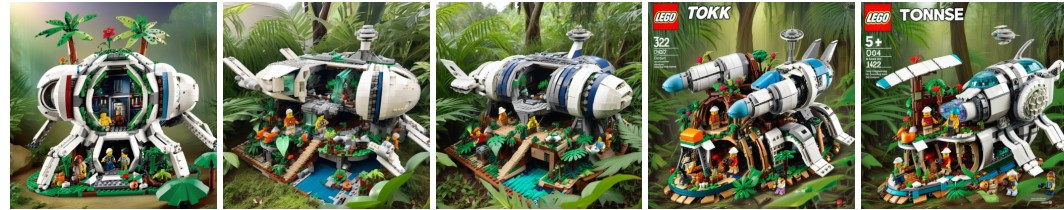

Prompt: a TOK lego set of an old steam train crossing a rickety bridge, in the style of TOK

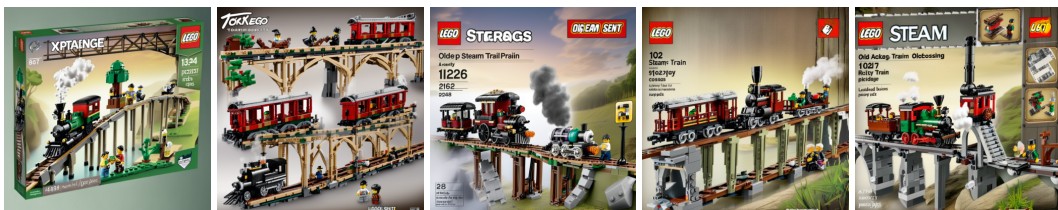

Prompt: a TOK lego set of a giant treehouse with rope bridges and zip lines, in the style of TOK

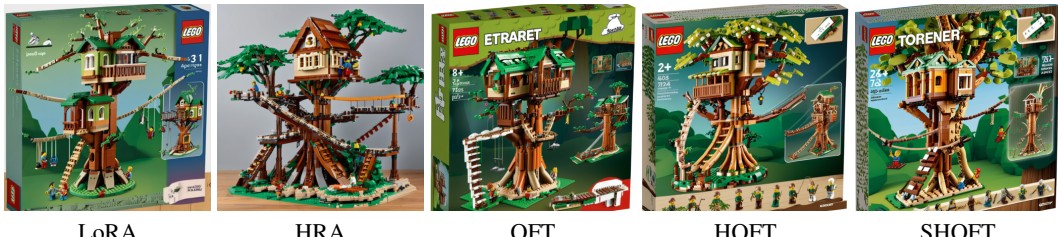

|      LoRA      |      HRA      |      OFT      |      HOFT      |      SHOFT      |

Figure 9: Comparison of different prompts in lego sets dataset

