# OpenReview forum: "HOFT: Householder Orthogonal Fine-tuning"
_NeurIPS.cc/2025/Conference — Submitted to NeurIPS 2025_

### Official Review · Reviewer_dReS · 2025-06-29

**Clarity:** 2
**Significance:** 3
**Originality:** 3
**Rating:** 4
**Confidence:** 3

**Summary:**

The paper introduces HOFT which is devised to conduct efficient orthogonal fine-tuning. The authors propose using two orthogonal matrices constructed via the CWY transform and efficient inverse approximations to adapt pre-trained weights effectively. Additionally, the paper introduces SHOFT which incorporates scaling transformations. Both methods are evaluated across commonsense reasoning, machine translation, subject-driven generation, and mathematical reasoning.

**Questions:**

* How well do HOFT and SHOFT scale to larger model sizes, and have the authors tested their effectiveness on models significantly beyond 14B parameters?


* Why are key orthogonal fine-tuning baselines such as OFT and BOFT missing from multiple evaluation tables (Tables 2, 3, 4, 6), and how would HOFT/SHOFT compare to them in terms of performance and computational cost?


* Could the authors clarify what the parameter ‘b’ represents in Table 1 and how it influences the complexity and performance of compared methods?


* What is the precise definition of the “coverage” metric in Table 1, and how is HOFT’s coverage calculated in practice?


* In Figure 8, both HOFT and SHOFT generate duck images with three chopsticks. could the authors comment on whether this indicates an out-of-distribution generation issue, and how well current metrics capture image qualities?

**Ethical Concerns:**

["NO or VERY MINOR ethics concerns only"]

**Final Justification:**

The authors have addressed most of my concerns in the rebuttal. I have therefore increased my score accordingly.

**Limitations:**

yes

**Paper Formatting Concerns:**

It would be great to provide more explicit methodological details on HOFT in the abstract for clarity.

**Quality:**

3

**Strengths And Weaknesses:**

### Strengths
* The theoretical framework behind using dual orthogonal matrices for fine-tuning is clearly articulated (e.g., CWY transform) .
* HOFT demonstrates improved computational complexity compared to existing orthogonal methods by leveraging parallelizable computations.

### Limitations
* The scalability and applicability of HOFT and SHOFT to different model sizes and large sizes are not thoroughly explored.
* The crucial baselines (OFT, BOFT) are missing in multiple experimental tables (Tables 2, 3, 6) which raises questions regarding comparative computation effectiveness.
* Missing explanations: Parameter 'b' in Table 1.
* Missing explanations: Coverage metric—its definition and significance, including the method of deriving HOFT coverage, require explicit clarification.
* Figure 8 (duck images with three chopsticks) suggests potential out-of-distribution issues not clearly addressed or explained in the main text.

---

> ### Author Rebuttal · Authors · 2025-07-31
>
> Thank you for your thoughtful review and insightful comments. We hereby address your concerns below:
>
> ---
>
> `Q1. How well do HOFT and SHOFT scale to larger model sizes, and have the authors tested their effectiveness on models significantly beyond 14B parameters?`
>
> Trying larger model sizes is of course really interesting. As suggested, to assess the effectiveness of HOFT and SHOFT on models significantly beyond 14B parameters, we run new experiments in the mathematical reasoning task with quantized Qwen2.5-32B and LLaMA3.1-70B. These are the results:
>
> ### Mathematical reasoning using quantized models (see Table 6):
>
> | Model            | Method  | Params (%) | GSM8K | Orca-Math |
> |------------------|---------|--------------:|------:|-----:|
> | **Qwen2.5-32B**  | QLoRA   |          0.07 |  68.0 | 77.4 |
> |                  | QDoRA   |          0.07 |  68.4 | 61.4 |
> |                  | QHOFT   |          0.07 |  71.8 | **80.6** |
> |                  | QSHOFT  |          0.07 | **75.0** | 66.9 |
> | **LLaMA3.1-70B** | QLoRA   |          0.12 |  76.4 | **82.7** |
> |                  | QDoRA   |          0.12 |  77.7 | OOM  |
> |                  | QHOFT   |          0.12 | **78.7** | 80.2 |
> |                  | QSHOFT  |          0.12 |  70.6 | 81.8 |
>
> As with LLaMA2-7B and LLaMA3.1-8B, QHOFT/QSHOFT achieve the best results for GSM8K with Qwen2.5-32B and LLaMA3.1-70B, and Orca-Math with Qwen2.5-32B. In the case of Orca-Math with LLaMA3.1-70B, QLoRA is slightly better than QHOFT/QSHOFT. In connection with this latter result, please note that we run an experiment at the limit of our computational resources, using a batch size of 1 with 4 accumulation steps, and thus we are not fully confident with the stability of the fine-tuning process.
>
> ---
>
> `Q2. Why are key orthogonal fine-tuning baselines such as OFT and BOFT missing from multiple evaluation tables (Tables 2, 3, 4, 6), and how would HOFT/SHOFT compare to them in terms of performance and computational cost?`
>
> We apologize for not providing all possible PEFT baselines. As suggested, we have carried out additional experiments following the same parameter settings as in previous experiments. Please see the results in the reply to Q1 by reviewer VCfn. In the case of Table 6, OFT and BOFT cannot be used with quantized models due to lack of practical implementations because of numerical constraints (related to the computation of matrix inverses with quantized weights).
>
> Regarding the HOFT/SHOFT computational cost, as compared to OFT and BOFT, we have run additional experiments in qualitative subject-driven generation with the following results:
>
> | Method  | Training time (hours) | Peak memory (GB) |
> |---------|----------------------:|-----------------:|
> | LoRA    |                   2.9 |             25.5 |
> | HRA     |                  35.8 |             25.7 |
> | OFT     |                   7.4 |             26.7 |
> | BOFT    |                  10.8 |             31.9 |
> | HOFT    |                   4.3 |             25.5 |
> | SHOFT   |                   4.3 |             25.5 |
>
>
> Clearly, HOFT/SHOFT largely outperform OFT and BOFT in terms of computational complexity.
>
> ---
>
> `Q3. Could the authors clarify what the parameter ‘b’ represents in Table 1 and how it influences the complexity and performance of compared methods?`
>
> Please accept our apologies for not clearly stating that ‘b’ denotes 'block size' in Table 1. As 'b' increases, the block diagonal orthogonal matrix is closer to a full orthogonal matrix. However, the increase of the block size also increases time complexity due to the cost of inverting b x b matrices.
>
> ---
>
> `Q4. What is the precise definition of the “coverage” metric in Table 1, and how is HOFT’s coverage calculated in practice?`
>
> You are right, we should have made us clearer as for the term 'coverage' in the paper. It does not refer to a metric, but a boolean mathematical property related to topology. In particular, OFT, BOFT, HRA and HOFT/SHOFT
> can be seen as different ways to parameterize the orthogonal group. We state that they cover (have a coverage over) the orthogonal group O(m) if the parameterization function is surjective. We are not aware of any technique to calculate the degree of coverage in practice.
>
> ---
>
> `Q5. In Figure 8, both HOFT and SHOFT generate duck images with three chopsticks. could the authors comment on whether this indicates an out-of-distribution generation issue, and how well current metrics capture image qualities?`
>
> Please note that LoRA also generates a duck image with three chopsticks. We do think this a mere example from which it is difficult to draw clear conclusions about the entire distribution.
>
> ---

---

> > ### Comment · Reviewer_dReS · 2025-08-05
> >
> > Thank you for the rebuttal and the additional experimental results. As most of my concerns have been addressed, I have updated my score accordingly.

---

### Official Review · Reviewer_VCfn · 2025-07-02

**Clarity:** 3
**Significance:** 2
**Originality:** 3
**Rating:** 4
**Confidence:** 3

**Summary:**

This paper proposes a PEFT method called Householder Orthogonal Fine-tuning (HOFT). Previous OFT methods have high runtime and memory costs, and they usually use a single orthogonal matrix for adaptation which limits the expressivity. To tackle these problems, HOFT updates two orthogonal matrices as direction components, and uses an approximation method to ensure efficiency. In addition, they propose Scaled Householder Orthogonal Fine-tuning to further match the learning dynamics of full fine-tuning. The authors conducted experiments in four distinct areas, including commonsense reasoning, machine translation, subject-driven generation and mathematical reasoning, where HOFT and SHOFT reach or exceed the performance of existing state-of-the-art PEFT baselines.

**Questions:**

See weaknesses.

**Ethical Concerns:**

["NO or VERY MINOR ethics concerns only"]

**Final Justification:**

The author response generally addresses all of my concerns and I updated my score to 4.

**Limitations:**

yes

**Quality:**

2

**Strengths And Weaknesses:**

**Strengths**

(1) This paper proposes a novel PEFT method: Householder Orthogonal Fine-tuning. Compared to previous orthogonal finetuning methods, HOFT updates two orthogonal matrices instead of one. In addition, the authors propose an approximation method to improve efficiency. The proposed method is novel and convincing.

(2) Some theoretical analysis is shown to demonstrate the advantages of HOFT that it better matches full finetuning dynamics.

(3) Extensive experiments on multiple tasks, including commonsense reasoning, machine translation, subject-driven generation, and mathematical reasoning, show HOFT outperforms baseline methods.

(4) The quantized version of the proposed method, QHOFT, outperformed the baseline methods as well.

**Weaknesses**

(1) The baselines used in different tasks seem arbitrary. For example, one important baseline OFT is only included in mathematical reasoning and  subject-driven generation, but not in commonsense reasoning and machine translation. DoRA also only appears in some of the tasks. What is the reason for this selection?  In my opinion, these PEFT methods should be generally applicable for any task.

(2) On commonsense reasoning and machine translation, the advantage of HOFT over LoRA is marginal. As HOFT leads to considerable training time and memory increase compared to LoRA, it is still more practical to directly use LoRA on these tasks. On subject-driven generation and mathematical reasoning tasks, the improvement over HRA baseline is also marginal, so the main advantage of HOFT here might be the efficiency compared to HRA. It is necessary to clearly state what the target scenario is for HOFT.

---

> ### Author Rebuttal · Authors · 2025-07-31
>
> Thank you for your thoughtful review and insightful comments. We hereby address your concerns below:
>
> ---
>
> `Q1. The baselines used in different tasks seem arbitrary. For example, one important baseline OFT is only included in mathematical reasoning and subject-driven generation, but not in commonsense reasoning and machine translation. DoRA also only appears in some of the tasks. What is the reason for this selection? In my opinion, these PEFT methods should be generally applicable for any task.`
>
> We apologize for not providing all possible PEFT baselines. As suggested, we have carried out additional experiments following the same parameter settings as in previous experiments:
>
> ### Commonsense reasoning (see Table 2):
>
> As shown in the table below, OFT baselines for LLama3.1-8B and Qwen2.5-7B are now provided. Unfortunately, experiments with larger models (Phi4-14B and Qwen2.5-14B) were not able to be fine-tuned at the current #Params setting due to surpassing H100 capabilities. In the case of BOFT, as it constructs a full orthogonal matrix, all models surpassed the H100 capabilities.
>
> | Model             | Method | #Params (%) | BoolQ | PIQA     | SIQA | HellaSwag | WinoGrande | ARC-e | ARC-c | OBQA | Avg. |
> |-------------------|--------|------------:|------:|---------:|-----:|----------:|-----------:|------:|------:|-----:|-----:|
> | **LLaMA3.1-8B**   | LoRA   |        0.35 |  88.2 | **88.5** | 80.3 |      96.7 |       80.5 |  91.9 |  82.3 | 87.4 | 87.0 |
> |     | DoRA   |        0.36 |  88.1 |    89.1  | 80.1 |      96.6 | **81.4**   |  92.0 |  82.5 | 86.8 | 87.1 |
> |    | OFT    |        0.35 |  87.8 |    87.9  | 79.4 |      95.7 |       77.4 |  92.6 |  82.8 | 88.2 | 86.5 |
> |        | HOFT   |        0.35 |  88.5 | **88.5** | **80.9** | **96.8** |       80.4 | **92.7** | **83.2** | **88.4** | **87.4** |
> |        | SHOFT  |        0.36 | **88.8** | **88.5** | 80.1 | **96.8** |       81.2 |  92.0 |  82.9 | 86.6 | 87.1 |
> | **Qwen2.5-7B**    | LoRA   |        0.35 |  88.4 |    89.5  | **79.6** | **96.8** | **82.5**   |  95.8 |  88.7 | 92.2 | 89.2 |
> |    | DoRA   |        0.36 |  88.9 | **89.8** | 79.2 | **96.8** | **82.5**   | **96.2** |  88.9 | 92.4 | **89.3** |
> |        | OFT    |        0.35 |  88.6 |    89.0  | 79.2 |      96.0 |       78.1 |  95.7 |  90.0 | 91.2 | 88.5 |
> |  | HOFT   |        0.35 | **89.0** |    89.1  | 79.2 |      96.4 |       80.4 |  95.9 |  88.4 | 92.4 | 88.9 |
> |              | SHOFT  |        0.36 |  88.8 |    89.5  | 79.5 |      96.5 |       80.7 |  95.7 | **89.1** | **93.4** | 89.2 |
>
>
> As observed, OFT does not provide as competitive results as HOFT/SHOFT.
>
> ### Machine translation (see Table 3):
>
> As requested, both OFT and BOFT baselines are now provided, including also average BLEU/COMET figures:
>
>
> | Model         | Method    | #Params (%) | Slovene BLEU | Slovene COMET | German BLEU | German COMET | Latvian BLEU | Latvian COMET | French BLEU | French COMET | Avg BLEU | Avg COMET |
> |---------------|-----------|------------:|-------------:|--------------:|------------:|-------------:|-------------:|--------------:|------------:|-------------:|---------:|----------:|
> | **NLLB‑3.3B** | Baseline  |           – |         39.7 |          87.5 |        39.3 |         86.2 |        31.2 |          81.3 |        38.5 |         84.9 |     37.2 |      85.0 |
> |               | LoRA      |        0.42 |         46.8 |          89.2 |        44.5 |     **87.7** |        38.2 |          83.9 |    **49.7** |     **87.8** |     44.8 |      87.2 |
> |               | DoRA      |        0.43 |         46.8 |          89.1 |    **44.7** |         87.6 |        38.2 |          83.9 |        49.5 |         87.7 |     44.8 |      87.1 |
> |               | OFT       |        0.43 |         45.5 |          89.1 |        44.2 |         87.5 |        36.9 |          83.3 |        49.1 |         87.6 |     43.9 |      86.9 |
> |               | BOFT      |      0.43 |      45.3 |       89.1 |     44.1 |      87.5 |        36.9 |          83.4 |        49.1 |         87.7 |     43.9 |      86.9 |
> |               | HOFT      |     0.42 |    **48.0** |          89.4 |        44.4 |     87.6 |    **38.6** |          83.9 |        49.5 |         87.7 | **45.1** |      87.2 |
> |               | SHOFT     |        0.43 |         46.4 |     **89.5** |        44.5 |     **87.7** |    **38.7** |   **84.0** |    **49.7** |     **87.8** |     44.8 | **87.3** |
> | **LLaMA2‑7B** | 0‑shot    |           – |         26.8 |          72.8 |        30.4 |   74.1 |         4.5 |    52.2 |        37.2 |         79.3 |     24.7 |      69.6 |
> |               | LoRA      |        0.19 |         39.3 |          84.7 |        41.5 |     86.9 |        15.5 |   66.2 |        47.0 |         87.2 |     35.8 |      81.3 |
> |               | DoRA      |        0.19 |         39.6 |          84.8 |        41.4 |      86.9 |    **16.2** |      **66.6** |        47.0 |         87.2 |     36.1 |      81.4 |
> |               | OFT       |        0.19 |         40.3 |          84.7 |        41.3 |      86.8 |        11.0 |          61.9 |        46.9 |         87.2 |     34.9 |      80.2 |
> |               | BOFT      |        0.19 |         40.4 |          84.9 |        41.2 |      86.8 |        11.0 |          61.6 |    46.9 |         87.2 |     34.9 |      80.1 |
> |               | HOFT      |        0.19 |         40.6 |          85.2 |        41.4 |         86.9 |        15.8 |      **66.6** |     47.0 |     **87.3** |     36.2 | **81.5** |
> |               | SHOFT     |        0.19 |    **41.2** |     **85.6** |    **41.6** |     **87.0** |        15.7 |      65.9 |    **47.1** |     **87.3** | **36.4** | **81.5** |
> | **LLaMA3.1‑8B** | 0‑shot  |           – |         34.2 |          77.8 |        40.9 |         86.2 |        22.9 |     70.8 |        41.6 |         82.7 |     34.9 |      79.4 |
> |               | LoRA      |        0.12 |         36.2 |       84.1 |        42.3 |         87.4 |        32.7 |      **80.9** |    **46.8** |         85.5 |     39.5 |      84.5 |
> |               | DoRA      |        0.12 |         42.4 |       85.0 |        42.2 |         87.4 |    **32.8** |          80.8 |        46.7 |         85.5 |     41.0 |      84.7 |
> |               | OFT       |        0.16 |         21.5 |          74.7 |        43.5 |         87.6 |        17.4 |          69.4 |        46.8 |         85.5 |     32.3 |      79.3 |
> |               | BOFT      |        0.16 |         22.1 |          75.9 |    **43.6** |         87.6 |        17.1 |          69.2 |        46.8 |         85.5 |     32.4 |      79.6 |
> |               | HOFT      |        0.12 |    **44.2** |     **86.6** |        42.9 |         87.5 |        32.2 |          80.4 |        46.7 |     **85.6** | **41.5** |  **85.0** |
> |               | SHOFT     |        0.12 |         43.6 |          86.4 |    **43.1** |     **87.7** |        31.9 |          80.4 |    **46.8** |     **85.6** |     41.4 |  **85.0** |
>
> As observed, HOFT/SHOFT are more competitive than OFT/BOFT.
>
> ### Subject-driven generation (see Table 4):
>
> As suggested, DoRA and BOFT baselines are now provided in the table below.
>
> | Method              | #Param (M) | DINO ↑ | CLIP-I ↑ | CLIP-T ↑ | LPIPS ↑ |
> |---------------------|-----------:|-------:|---------:|---------:|--------:|
> | Real Images         |      —     |  0.764 |    0.890|    —     |  0.562  |
> | BOFT b=8, m=2   |     89.5   |  0.545  |   0.735  |  0.258   | **0.785** |
> | DoRA                |      0.81  |  0.633 |    0.781|    0.246 |  0.763  |
> | HOFT r=2  |      0.40  |  0.657 |    0.793|    0.239 |  0.758  |
> | SHOFT r=2|      0.41  |  0.658 |    0.793|    0.241 |  0.757  |
> | HOFTr=4  |      0.80  | **0.680**  |    **0.810**|    0.235 |  0.752  |
> | SHOFT r=4 |      0.81  | **0.680**  |    0.808|    0.235 |  0.747  |
>
> As shown above, DoRA obtains better results than its counterpart LoRA, but it is not as competitive as HOFT/SHOFT. However, BOFT required an increased number of parameters to obtain competitive results compared with the rest of PEFTs.
>
> ### Mathematical reasoning (see Table 5):
>
> As requested, DoRA baseline is now reported below.
>
> | Method | GSM8K | MATH |
> |--------|------:|-----:|
> | DoRA   |  53.5 |  9.1 |
> | HOFT   | **56.6** | 8.9 |
> | SHOFT  |  55.0 | **9.8** |
>
> As shown in the table above, DoRA obtains better performance than PiSSA and LoRA but is not as competitive as HOFT/SHOFT.
>
> ---
>
> `Q2a. On commonsense reasoning and machine translation, the advantage of HOFT over LoRA is marginal. As HOFT leads to considerable training time and memory increase compared to LoRA, it is still more practical to directly use LoRA on these tasks.`
>
> In commonsense reasoning and machine translation, HOFT/SHOFT consistently surpass the rest of PEFT methods. We also performed an ablation study on the training set size in the commonsense reasoning tasks, confirming this superiority (see reply to reviewer QX2R).
>
> We added two columns to the machine translation table above: average BLEU and COMET, to highlight HOFT/SHOFT’s consistent improvements. The best average BLEU/COMET are always achieved by HOFT/SHOFT.
>
> Compared with SOTA PEFT methods (i.e., LoRA), we are convinced the empirical evidence clearly supports our claim that HOFT/SHOFT consistently show superior performance across all tasks for a marginal computational cost increase over LoRA.
>
> `Q2b. On subject-driven generation and mathematical reasoning tasks, the improvement over HRA baseline is also marginal, so the main advantage of HOFT here might be efficiency compared to HRA. It is necessary to clearly state what the target scenario is for HOFT.`
>
> It is true that on subject-driven generation, HOFT/SHOFT achieve a modest improvement over HRA. Nevertheless, we believe HOFT/SHOFT are consistently superior to SOTA PEFT methods, especially given HRA’s training slowdowns.
>
> In summary, HOFT/SHOFT are very competitive rivals to SOTA PEFT methods across the usual target scenarios.

---

> > ### Comment · Reviewer_VCfn · 2025-08-05
> >
> > Thank you for the response, which addresses most of my concerns. I have therefore increased my score.

---

### Official Review · Reviewer_qX2R · 2025-07-05

**Clarity:** 2
**Significance:** 2
**Originality:** 3
**Rating:** 4
**Confidence:** 3

**Summary:**

This paper introduces HOFT and SHOFT, both of which are novel parameter-efficient fine-tuning (PEFT) approaches leveraging two orthogonal transformations (pre- and post-multiplication) parameterized by accumulated Householder reflections with a CWY transform and an approximate inverse. The rationale for their proposed method is to preserve the geometric properties of the pre-trained weights and reducing the computational overhead of similar methods like OFT and HRA.

**Questions:**

- How does HOFT/SHOFT compares with other methods on (visual) instruction tuning eval?
- How does HOFT/SHOFT performs with different training set size?

**Ethical Concerns:**

["NO or VERY MINOR ethics concerns only"]

**Limitations:**

- See Weaknesses.

**Quality:**

2

**Strengths And Weaknesses:**

Strengths
- The paper is generally well-written.
- Clear mathematical justification for using two orthogonal matrices to capture all distance-preserving transformations.
- The authors provide multiple qualitative results demonstrating that HOFT/SHOFT methods perform competitively with other methods.

Weaknesses
- No results for instruction tuning evaluations or visual instruction tuning.
- No ablation on training data size.

---

> ### Author Rebuttal · Authors · 2025-07-31
>
> Thank you for your thoughtful review and insightful comments. We hereby address your concerns below:
>
> ---
>
> `Q1. How does HOFT/SHOFT compares with other methods on (visual) instruction tuning eval?`
>
> Thank you for raising this important point. We agree that assessing HOFT/SHOFT on (visual) instruction tasks is interesting, and as pointed in Section 6 of our manuscript, a comprehensive evaluation of HOFT/SHOFT under visual instruction‐tuning benchmarks is foreseen as a future work. Unfortunately, time and computing limitations did not allow us to include results on these tasks for the current work. We tried to get results before the rebuttal deadline, but we did not manage to complete the experiments due to computational limitations.
>
> ---
>
> `Q2. How does HOFT/SHOFT performs with different training set size?`
>
> Thank you for your excellent question. As suggested, to assess the robustness of HOFT and SHOFT under an increasing amount of training data, we have performed an ablation study on LLaMA3.1‑8B using 5%, 10%, 25%, 50% and 100% of the commonsense reasoning training split. The results are summarized below:
>
> | **Method** | **5%** | **10%** | **25%** | **50%** | **100%** |
> |------------|-------:|--------:|--------:|--------:|---------:|
> | LoRA       |   81.8 |    83.5 |    85.0 |    86.4 |     87.0 |
> | DoRA       |   81.7 |    83.4 |    84.9 | **86.5** |     87.1 |
> | OFT        |   77.8 |    80.7 |    83.2 |    85.0 |     86.5 |
> | HOFT   |   82.0 | **83.7** |    85.1 | **86.5** | **87.4** |
> | SHOFT  | **82.3** |    83.6 | **85.4** |    86.3 |     87.1 |
>
> As shown, both HOFT and SHOFT consistently surpass other PEFT methods across all data regimes, with SHOFT leading at the lowest-data setting (5%) and HOFT achieving the highest accuracy at the full-data scale (100%). We will include the complete experimental protocol, hyperparameter settings, and detailed results in the revised manuscript.
>
> ---

---

> > ### Comment · Reviewer_qX2R · 2025-08-04
> > **Rebuttal Response**
> >
> > Thank you for the response. I will keep my score as 4: Borderline Accept.

---

### Decision · Program_Chairs · 2025-09-17

**Decision:**

Reject

**Comment:**

The authors have provided additional numerical results in response to the reviewers' comments on the deficiencies of the numerical experiments in the manuscript. However, the scope of the additional results suggests that the manuscript will need to undergo a major revision to bring it up to standard. This is also reflected in the borderline ratings of all three reviewers. In addition, as pointed out by one of the reviewers, there should be a more thorough discussion on the target scenario of the proposed approach. In view of the above, I regrettably have to reject the manuscript.